# The Influence of SiO_2_ + SiC + Al (H_2_PO_4_)_3_ Coating on Mechanical and Dielectric Properties for SiC_f_/MWCNTS/AlPO_4_ Composites

**DOI:** 10.3390/ma15155178

**Published:** 2022-07-26

**Authors:** Yan Zhu, Feng Wan, Jianhui Yan, Hongmei Xu

**Affiliations:** 1School of Materials Science and Engineering, Hunan University of Science and Technology, Xiangtan 411201, China; 2020060428@mail.hnust.edu.cn (Y.Z.); jhyan@hnust.edu.cn (J.Y.); xhmhnust@163.com (H.X.); 2Hunan Provincial Key Laboratory of Advanced Materials for New Energy Storage and Conversion, Hunan University of Science and Technology, Xiangtan 411201, China

**Keywords:** SiC fibers, oxidation, SiP_2_O_7_, solid solution

## Abstract

SiC fiber-reinforced AlPO_4_ matrix (SiC_f_/MWCNTs/AlPO_4_) composites were fabricated using a hot laminating process with multi-walled carbon nanotubes (MWCNTs) as the absorber. A coating prepared from SiO_2_ + SiC + Al (H_2_PO_4_)_3_ was applied to the surface of the SiC_f_/MWCNTs/AlPO_4_ composites prior to an anti-oxidation test at 1273 K in air for 40 h. The anti-oxidation effect was verified by a three-point bending test, scanning electron microscopy, transmission electron microscopy, X-ray diffraction, and a dielectric property test. Anti-oxidation mechanism investigations revealed that the coating effectiveness could be attributed to three substances, i.e., SiO_2_, SiP_2_O_7,_ and SiO_2_ + AlPO_4_ solid solution from the reactions of SiC + O_2_→SiO_2_ + CO, SiO_2_ + P_2_O_5_→SiP_2_O_7_ and SiO_2_ + AlPO_4_→solid solution, respectively.

## 1. Introduction

Electromagnetic wave-absorbing materials, designed to decrease reflected electromagnetic radiation by absorbing electromagnetic waves and transforming it into other energy, is a topic of extensive interest in the aerospace and military fields [1,2,3,4,5,6]. Currently, the research for applications in high-temperature environments is the main topic for electromagnetic wave absorbing materials.

Continuous fiber-reinforced ceramic matrix composites (SiC_f_/SiC [7], SiC_f_/C [8], C_f_/C [9], C_f_/SiC [10], and SiC_f_/AlPO_4_ [11]), showing excellent fracture toughness, good thermal stability, and environmental durability, have been evaluated and modified for use as structural electromagnetic wave-absorbing materials. Especially, continuous SiC fiber-reinforced AlPO_4_ (SiC_f_/AlPO_4_) composites have demonstrated good potential [12,13,14]. Their low dielectric constants provide the opportunity to tailor the dielectric properties and wave absorbing abilities by the addition of conductive fillers (carbon black, carbon nanotube, and graphene). However, the conductive fillers, SiC fibers, and C fibers are easily oxidized in oxidation environments, which limits the high-temperature application for ceramic matrix composites. Therefore, it is very important to form high-temperature antioxidant coatings on the surfaces of ceramic matrix composites to improve their oxidation resistance.

Currently, many efforts have been made regarding antioxidant coatings [15,16,17]. SiC ceramic coatings are usually used as bonding layers in environmental barrier coating (EBC) for C/SiC composite coatings owing to their good chemical and physical compatibilities with C/SiC composites [18,19]. However, micro-cracks develop owing to the difference in the thermal expansion coefficient between the SiC coating and C/SiC composites during the long oxidation process. To avoid the above problems, top coatings should be prepared on the surface of the SiC coating to heal the micro-cracks, such as mullite, MoSi_2_, ZrSiO_4_, Y_2_SiO_5_, CrSi_2_, and c-AlPO_4_ top coatings. AlPO_4_ ceramic, with many excellent properties such as a high melting point (above 1773 K), strong self-healing ability, low Young’s modulus, and low oxygen permeability is usually an ideal high-temperature anti-oxidation coating material for many ceramic composites [20,21]. With the lowest low oxygen permeability in various oxide ceramics, SiO_2_ ceramic also shows its advantage in the field of anti-oxidation coating [22,23].

Based on these results, a multi-composition coating, including SiC, SiO_2_, and AlPO_4_ was prepared on the surface of SiC_f_/MWCNTs/AlPO_4_ composites in this paper. The anti-oxidation efficiency of the coating was proved and examined in detail.

## 2. Experimental Details

### 2.1. Materials

The SiC fiber was provided by the National University of Defense Technology (Changsha, China). The 2D SiC fiber cloths with a 40% fiber volume fraction were fabricated by Nanjing Glass Fiber Institute (Nanjing, China). MWCNTs used as conductive filler were supplied by the Shenzhen Nanotech port Co. Ltd., Shenzhen, China. The diameter of MWCNTs ranged from 20 to 80 nm, and length was 5–15 μm, and the purity was 95%. Figure 1a,b show the SEM image of 50 vol.% 2D SiC cloths and the TEM image of MWCNTs, respectively. The diameters of SiO_2_, Al_2_O_3_, and β-SiC powders are in the range of 1–5 μm.

### 2.2. Preparation of the Composites

The Al(H_2_PO_4_)_3_ solution, which is a precursor of AlPO_4_, was synthesized from aluminum hydroxide (Al(OH)_3_) and orthophosphoric acid (H_3_PO_4_, 85%). Al(OH)_3_ at 1 mol was dispersed in deionized water, and H_3_PO_4_ (85%) at 3 mol was added into the suspension liquid to maintain the theoretical Al/P atomic ratio of 1:3. The mixed solution was then allowed to react at 90 °C for several hours, and the viscous Al(H_2_PO_4_)_3_ solution was obtained. The MWCNTs and Al_2_O_3_ powders were uniformly mixed with as-received Al(H_2_PO_4_)_3_ solution by ball milling for 4 h to obtain the slurry. The SiC fiber cloths were impregnated in the slurry. After air drying for 24 h, the 10 sheets of cloths obtained were laminated and hot pressed in a steel die at 100 and 200 °C for 1 h in turn. A pressure of 3 MPa was applied when the temperature reached 100 °C, and such pressure was maintained until the end of hot pressing. Then, these samples were heated at a rate of 5 °C/min in a vacuum furnace to 500 °C for 1 h, and SiC_f_/MWCNTs/AlPO_4_ composites were obtained.

### 2.3. Preparation of the Coating

Al(H_2_PO_4_)_3_ solution was mixed with the SiO_2_ and SiC powders in the ratio 5:3:2 (*w*/*w*) Al(H_2_PO_4_)_3_:SiO_2_:SiC. After ball milling for 3 h, the obtained mixture was brushed onto the surface of SiC_f_/AlPO_4_ composites and dried at 373 K for 1 h prior to annealing at 1473 K for 3 h at a heating rate of 283 K/min in vacuum atmosphere. After cooling at ambient temperature, the sample was given two infiltration–drying–annealing cycles to yield the coated SiC_f_/AlPO_4_ composites. Uncoated and coated SiC_f_/AlPO_4_ composites were heated to 1273 K in a muffle furnace. Treated samples were cooled to room temperature under ambient conditions.

### 2.4. Test Equipment

Morphology and microstructure were characterized by SEM (ZEISS Supra 55, Mainz, Germany) and TEM (G-20, FEI-Tecnai, Hillsboro, OR, USA). Phase evolution characterizations were determined by X-ray diffraction (XRD; X’Pert Pro, Philips, Amsterdam, The Netherlands).

The flexural strength of composites at room temperature was obtained by the three-point bending test, with a crosshead rate of 0.5 mm/min and an outer support span of 30 mm. The test was conducted following the general guidelines of ASTM standard C1341.

The complex permittivity values for the composites were measured based on the measurements of the reflection and transmission module between 8.2 and 12.4 GHz. The method was performed in the fundamental wave-guide mode TE10 using rectangular samples (10.16 mm × 22.86 mm × 3.00 mm). After calibration using an intermediate of a short circuit and blank holder, the reflection and transmission coefficients were obtained using an automated measuring system (E8362Bnetworkanalyzer). For dielectric materials (μ0 = 1, μ″ = 0), the relative error varied between 1% (pure dielectric) and 10% (highly conductive material). The schematic diagram is shown in Figure 2.

## 3. Results and Discussion

### 3.1. Investigation of Bending Strength

The bending strengths for SiC_f_/MWCNTs/AlPO_4_ composites obtained by the three-point bending test are reflected in Figure 3.

The three specimens initially showed an elastic response with increasing displacement. After reaching the maximum strength, the bending strength of the as-received specimen displayed an inelastic decrease before reducing abruptly. This differed from the curves of the oxidized specimens (with and without the coating), which showed a direct reduction in bending strength at maximum strength. The brittle fracture for the three curves could be attributed to the absence of an interface, which led to the loss of toughening mechanisms including fiber pull-out and debonding, and crack deflection. After oxidizing for 40 h, the bending strength of the coated specimen decreased from 205 to 190 MPa, and the displacement was reduced to 0.38 mm. These effects were due to the influence of high temperature on the SiC fibers. The specimen without the coating attained a bending strength of 60 MPa and displacement of 0.14 mm.

The corresponding fracture surface morphologies of the specimens (with and without the coating) subjected to oxidizing conditions are given in Figure 4. The fracture surface of the coated specimen was smooth and little fiber pull-out was observed (Figure 4a). The cross sections of SiC fibers were complete and clearly visible. Figure 4b,c show the SEM and TEM pictures of uncoated SiC_f_/MWCNTs/AlPO_4_ composites undergoing 40 h oxidation. Obviously, a strong bond occurred and a reaction zone was formed between the AlPO_4_ matrix and SiC fiber, which could be attributed to the reaction of the AlPO_4_ matrix and SiO_2_, produced from the oxidation of SiC.

### 3.2. Investigation of Dielectric Property

The real part (ε′) and imaginary part (ε″) of the complex permittivity for the coated SiC_f_/AlPO_4_ composites are shown in Figure 5. Notably, Sample 1 denotes the coated SiC_f_/AlPO_4_ composites without MWCNTs added. Sample 2 denotes the coated SiC_f_/AlPO_4_ composites with 1.5 wt.% MWCNTs added. Sample 3 denotes the coated SiC_f_/AlPO_4_ composites with 1.5 wt.% MWCNTs added, which undergo 40 h oxidization.

The complex permittivity values of pure SiC_f_/AlPO_4_ composites (no coating and MWCNTs) have been discussed in [14]. The value of ε′ was in the range of 3.6–4.1 and the value of ε″ was in the range of 0.1–0.2. The values of ε′ and ε″ were small due to the insulated AlPO_4_ matrix, which was observed in Table 1. After the introduction of the coating, the ε′ and ε″ values for the SiC_f_/AlPO_4_ composites were in the range of 4.2–4.5 and 0.2–0.5 within the entire X-band, respectively. Compared to the result of Sample 1, the values of ε′ and ε″ showed little change, which proved that the introduction of the coating had little influence on the dielectric property. This was ascribed to the low dielectric constants of the coating substances, reflected in Table 1. With the introduction of 1.5 wt.% MWCNTs, the ε′ and ε″ for the coated SiC_f_/MWCNTS/AlPO_4_ composites ranges increased from 4.2–4.5 to 5.0–6.3 and 0.2–0.5 to 1.8–3.6, respectively. The main reasons can be given next.

Complex permittivity is expressed by the following equation: ε = ε′ − jε″. ε′ is an expression of the polarization ability of a material. ε″ is an expression of the capacity of dielectric losses, which comprise polarization loss and electric conductance loss. The complex permittivity affects the absorbing wave property. When the value of ε′ is too high, the electromagnetic wave cannot enter the composites, leading to a poor absorbing wave effect. When the value of ε″ is too small, the electromagnetic wave cannot be consumed, leading to a poor absorbing wave effect. So, a suit value of ε″/ε′ is needed to satisfy the impedance matching rule.

According to the Debye theory of the dielectric, ε′ and ε″ of the composites can be calculated as follows:(1)$$\varepsilon^{\prime }{=\varepsilon }_{\infty }+\frac{\varepsilon_{s}{-\varepsilon }_{\infty }}{{1+\omega }^{2}\tau^{2}}$$
(2)$$\varepsilon^{\prime \prime }=\frac{{(\varepsilon }_{s}{-\varepsilon }_{\infty })\omega \tau (T)}{{1+\omega }^{2}{\tau (T)}^{2}}+\frac{\sigma (T)}{{\omega \varepsilon }_{0}}$$
where ε_s_ is the static permittivity, ε_∞_ is the permittivity at the high-frequency limit, ω is the angular frequency, τ is the relaxation time, σ(T) is the temperature-dependence electrical conductivity, and ε_0_ is the dielectric constant in a vacuum.

As described in Formulas (1) and (2), ε′ was determined by the relaxation time (τ). ε″ was determined by both the relaxation time (τ) and electrical conductivity of the composites (σ(T)). The possible polarization mechanisms at the microwave frequency included electronic, atomic, relaxation, and space charge polarizations. The contribution of atomic and electronic polarizations to permittivity was small and negligible. The effect of space charge polarization on the GHz range was lost because a long duration of time was required to establish polarization. So, the increase in ε′ could be attributed to the electronic relaxation polarization enhanced by the MWCNTs. The introduction of MWCNTs not only brought the electronic relaxation polarization, but also made the electrical conductivity of the composites increase by free electrons shifting and hopping, which explains the increase in ε″.

After 40 h oxidization, the values of ε′ and ε″ for the coated SiC_f_/AlPO_4_ composites with 1.5 wt.% MWCNTs showed little change compared with the values before oxidation. These results showed that the MWCNTs were still present and functional. These findings demonstrated that the anti-oxidation effect of the coating was effective for the SiC_f_/MWCNTs/AlPO_4_ composites in an oxidizing environment at 1273 K.

### 3.3. Investigation of the Coating

Figure 6a shows the fracture surface image of coated SiC_f_/MWCNTs/AlPO_4_ composites before oxidization. The coating showed a strong bond with the AlPO_4_ matrix, and no obvious boundary was distinguished (indicated by the black arrows). At the same time, fiber pull-out was observed in the image, which proved that SiC fibers had no reaction with the AlPO_4_ matrix and the coating was effective. Figure 6b shows the surface image of the coating after preparation. It was observed from Figure 6b that the coating was dense and smooth, which showed a glassy state. No holes and cracks existed.

According to the phase diagram of AlPO_4_-SiO_2_ [23], some phases of solid solution (C-AlPO_4_ solid solution, T-AlPO_4_ solid solution, Cr- SiO_2_ solid solution, and Tr- SiO_2_ solid solution) might be formed when the preparation temperature of the coating was maintained at 1273 K, and these were dependent on the content of AlPO_4_ and SiO_2_ in the mixture. Consequently, this result was theoretically responsible for the strong bond between the AlPO_4_ matrix and the coating.

The XRD spectrum of reaction products derived from the Al(H_2_PO_4_)_3_ solution and SiO_2_ is reflected in Figure 7. Four major peaks around 2θ values of 20.4°, 24.2°, 25.9°, and 30.4° (at 10 wt.% SiO_2_) were homologous with the crystal phase of Al(PO_3_)_3_ and decreased in intensity with the increasing content of SiO_2_ and were absent at 40 wt.% SiO_2_. However, the intensity of the two peaks around 2θ values of 20.8° and 26.5° increased. Supposing these peaks were ensured to be SiO_2_, the disappearance of Al(PO_3_)_3_ peaks could not be accepted; if they are ensured to be AlPO_4_ (i.e., decomposition products of Al(PO_3_)_3_), the disappearance of SiO_2_ peaks could not be accepted. Hence, it was concluded that the SiO_2_ reacted with AlPO_4_ to form a solid solution, which was responsible for the diffraction peaks in the XRD spectrum of 40 wt.% SiO_2_. The continuous decomposition and final exhaustion for Al(PO_3_)_3_ was attributed to the consumption of AlPO_4_ from an abundance of SiO_2_. Hence the results given in Figure 4b and Figure 6b experimentally confirm that the SiO_2_ -AlPO_4_ solid solution was tightly correlated with the strong bond between the AlPO_4_ matrix and the coating.

The presence of low melting point SiP_2_O_7_, derived from the reaction of SiO_2_ and P_2_O_5_ (Al(PO_3_)_3_→AlPO_4_+ P_2_O_5_), also strengthened the bonding within the coating and filled the holes and cracks in the coating to make it be a smooth, dense, and glassy state. This could be confirmed by the phase diagram of SiO_2_-P_2_O_5_ [14]. These two chemical reactions contributed to the formation of a dense coating.

Figure 8a shows the fracture surface image of coated SiC_f_/MWCNTs/AlPO_4_ composites after 40 h oxidization. The reaction of the AlPO_4_ matrix and the SiO_2_ filler strengthened the bond between the matrix and coating. Figure 8b shows the surface image of the coating after 40 h oxidization. It was observed from Figure 8 that the coating was still dense and smooth, which showed a glassier state. No holes and cracks existed due to the fill of low melting point SiP_2_O_7_. On one hand, the routes of oxygen diffusion were filled due to the absence of holes and cracks; on the other hand, the efficiency of oxygen diffusion was decreased due to the SiO_2_, which was composed of the raw materials SiO_2_ and reaction SiO_2_ from the oxidization of SiC. These results proved that the coating was effective in preventing composite oxidization.

Table 2 shows the results of bend strength, dielectric constants. Obviously, it can be found that the coating is effective in protecting the SiC_f_/MWCNTS/AlPO_4_ composites from being oxidized.

Figure 9 shows the schematic diagram of the anti-oxidation mechanism for the coating. During the preparation of the coating, P_2_O_5_ (g) is readily released while the formation of SiP_2_O_7_ slows. However, the relatively long oxidation time was enough for P_2_O_5_ to react with SiO_2_ and form SiP_2_O_7_. Figure 9 gives a summary schematic representation of the mechanism of anti-oxidation based on the results from this study. During the preparation of the coating, the AlPO_4_ matrix reacts with SiO_2_ to form a SiO_2_ -AlPO_4_ solid solution leading to a strong chemical bond between SiC_f_/AlPO_4_ composites and the coating. The formation of SiP_2_O_7_ and SiO_2_ -AlPO_4_ solid solution facilitates the bonding of the particles in the coating, which contributes to the formation of a dense coating. Under oxidizing conditions, the SiC in the coating is partially transformed into SiO_2_ as it consumes the incoming oxygen gas, and the decomposition of Al(PO_3_)_3_ increases with the increasing time. The reactions of SiO_2_ -AlPO_4_ and SiO_2_ -P_2_O_5_ occur throughout the coating, linking particles to form a dense coating of low oxygen permeability.

Table 3 shows the calculated Gibbs free energy (∆G) for the oxidation of SiC at 1273 K. The reaction shown by Code (4) was favored by its minimal value of ∆G. As oxygen gas is introduced, SiC particles are oxidized to SiO_2_, which then reacts with P_2_O_5_ and AlPO_4_. The integration of SiO_2_, SiP_2_O_7_, and the SiO_2_ -AlPO_4_ solid solution into the coating is effective in preventing the oxygen gas from further diffusion into the SiC_f_/MWCNTs/AlPO_4_ composites.

## 4. Conclusions

This study presents a detailed investigation of the anti-oxidation mechanism of the SiO_2_ + SiC + Al(H_2_PO_4_)_3_ coating. The anti-oxidation effect of the SiC_f_/MWCNTs/AlPO_4_ composites in an oxidizing environment (1273 K, 40 h) was confirmed by a three-point bending test, microstructure characterization, and dielectric property. SiC_f_/MWCNTs/AlPO_4_ composites were chemically bonded with the coating. Oxygen gas in the environment was consumed by SiC particles to form SiO_2_, which subsequently reacted with P_2_O_5_ and AlPO_4_ to form SiP_2_O_7_ and SiO_2_ -AlPO_4_ solid solution, respectively. The integration of SiO_2_, SiP_2_O_7_, and SiO_2_ -AlPO_4_ solid solution into the coating was effective in preventing the oxygen gas from consuming MWCNTs. The coating gives SiC_f_/MWCNTs/AlPO_4_ composites the potential to be applied as high-temperature structural wave absorbing materials.

## Figures and Tables

**Figure 1 materials-15-05178-f001:**
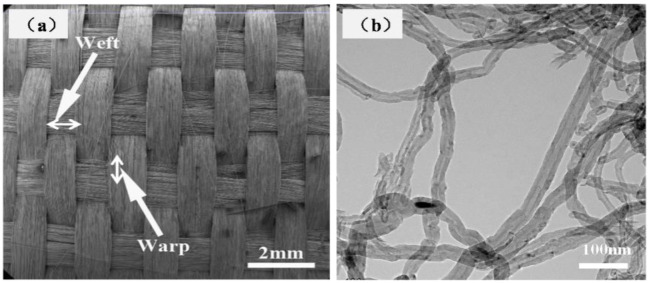
(**a**) 2D SiC fibers cloth and (**b**) TEM image of MWCNTs.

**Figure 2 materials-15-05178-f002:**
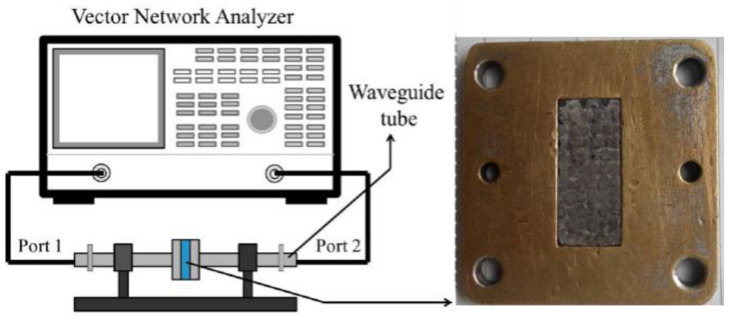
Schematic diagram for the dielectric property measurement.

**Figure 3 materials-15-05178-f003:**
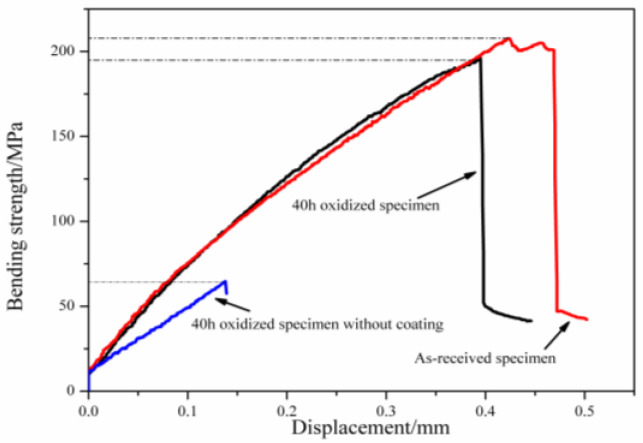
Typical stress–displacement curves of SiC_f_/MWCNTs/AlPO_4_ composites.

**Figure 4 materials-15-05178-f004:**
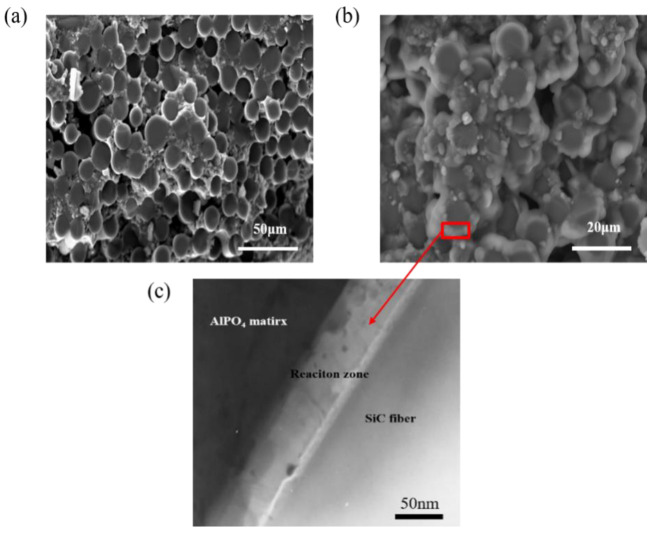
(**a**) SEM image of coated SiC_f_/MWCNTs/AlPO_4_ composites after oxidization, (**b**) SEM images of uncoated SiC_f_/MWCNTs/AlPO_4_ composites after oxidization, (**c**) TEM images of uncoated SiC_f_/MWCNTs/AlPO_4_ composites after oxidization.

**Figure 5 materials-15-05178-f005:**
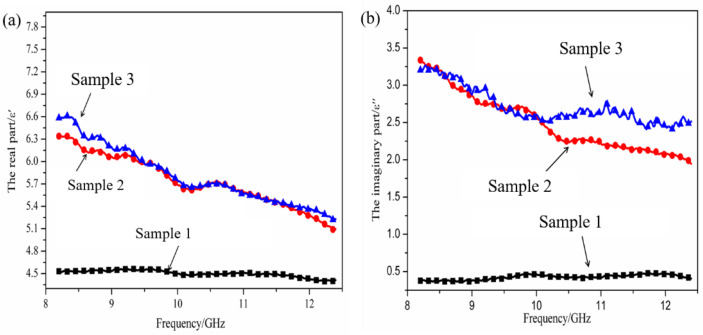
The complex permittivity values for the coated SiC_f_/MWCNTs/AlPO_4_ composites: (**a**) the real part (ε′), (**b**) the imaginary part (ε″).

**Figure 6 materials-15-05178-f006:**
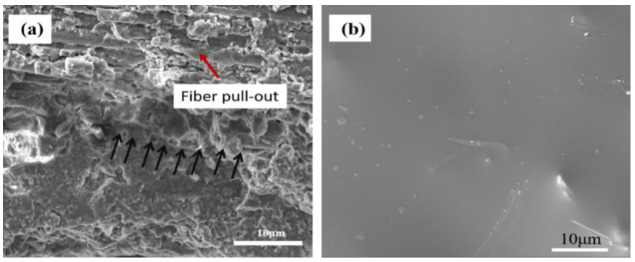
(**a**) The fracture surface image of coated SiC_f_/MWCNTs/AlPO_4_ composites and (**b**) the surface image of the coating before oxidization.

**Figure 7 materials-15-05178-f007:**
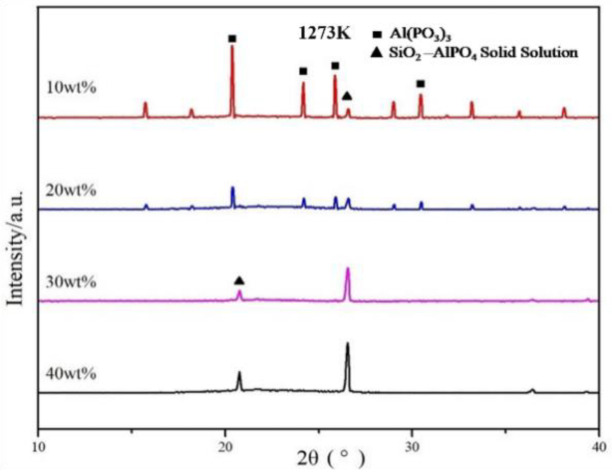
The XRD spectrum of reaction products derived from Al(H_2_PO_4_)_3_ solution and SiO_2_.

**Figure 8 materials-15-05178-f008:**
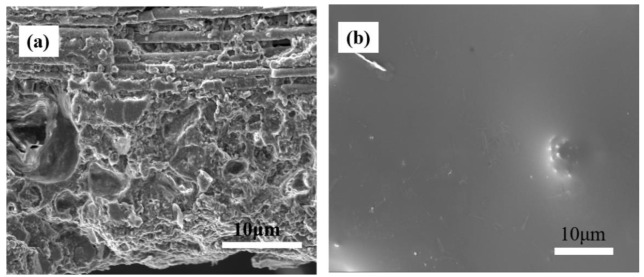
(**a**) The fracture surface image of coated SiC_f_/MWCNTs/AlPO_4_ composites after 40 h oxidization and (**b**) the surface image of the coating after oxidization.

**Figure 9 materials-15-05178-f009:**
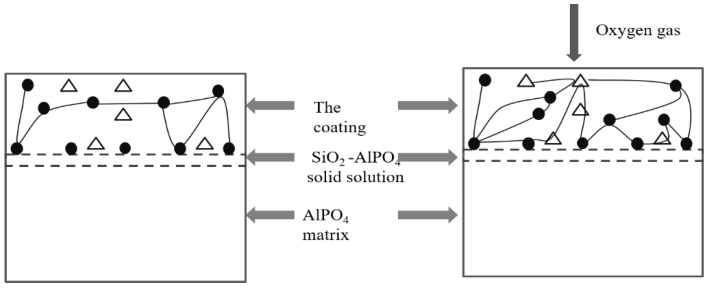
The schematic diagram of anti-oxidation mechanism for the coating (•—AlPO_4_, Δ —SiC).

**Table 1 materials-15-05178-t001:** The dielectric constants of several substances.

	Substance	AlPO_4_	SiO_2_-AlPO_4_ Solid Solution	SiP_2_O_7_	SiO_2_
Dielectric Constant					
ε′		4.0–4.3	3.8–4.2	2.0–2.4	3.4–3.7
ε″		0.1–0.3	0.1–0.3	0.1–0.2	0.2–0.4

**Table 2 materials-15-05178-t002:** The results of bend strength, dielectric constants.

	Bend Strength/MPa	ε′	ε″
Pure SiC_f_/AlPO_4_ composites		3.6–4.1	0.1–0.2
Coated SiC_f_/AlPO_4_ composites		4.2–4.5	0.2–0.5
Coated SiC_f_/MWCNTS/AlPO_4_ composites		5.0–6.3	1.8–3.6
Coated SiC_f_/MWCNTS/AlPO_4_ composites of oxidization		5.2–6.6	2.3–3.2
Coated SiC_f_/MWCNTS/AlPO_4_ composites	205		
Coated SiC_f_/MWCNTS/AlPO_4_ composites of oxidization	190		
Uncoated SiC_f_/MWCNTS/AlPO_4_ composites of oxidization	60		

**Table 3 materials-15-05178-t003:** The ∆G of SiC oxidation.

Code	Reaction	∆G/kJ/mol at 1273 K
(1)	SiC + 1/2O_2_ ⇔ SiO(g) + C	−156.07
(2)	SiC + O_2_ ⇔ SiO_2_ + C	−632.91
(3)	SiC + O_2_ ⇔ SiO(g) + CO(g)	−379.94
(4)	SiC + 3/2O_2_ ⇔ SiO_2_ + CO(g)	−855.27
(5)	C + 1/2O_2_ ⇔ CO(g)	−223.59

## Data Availability

Not applicable.

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
