# Peer review of "The Influence of SiO2 + SiC + Al (H2PO4)3 Coating on Mechanical and Dielectric Properties for SiCf/MWCNTS/AlPO4 Composites"

_materials, 2022, doi:10.3390/ma15155178_

Round 1

Reviewer 1 Report

Authors have to incorporate following in the manuscript:

1. Introduction part of the manuscript should be re-written by highlighting the  recent research work on the coating.

2. Authors should convert the manuscript in the following manner Introduction, Methodology or experimental details, materials used, Results and discussion, conclusions.

3. The manuscript should contain the preparation of composites in a systematic manner.

4. How coating process developed and materials selected should be covered in the experimental details.

5. Abstract contain the Multi walled carbon nano tubes, but title of the manuscript covered only SiC fibers.

6. All the test parameters as per ASTM standards should be explained in the manuscript.

7. A through results and discussion section is required in the manuscript.

Reviewer 2 Report

Dear authors,

Although the idea is very good, the work is too short. It is not clear how the nanotubes are introduced, what their function is. Only one experiment is done.

In the world of carbon it is well known that the presence of phosphoric acid and its derivatives retard oxidation, therefore it is not new. The formation of mixed oxides with silica has also been reported, albeit with another compound [1].

The system would be similar because it tries to protect SiC/C composites.

To be published, in my humble opinion, more experiments should be done at other temperatures, to verify that they really have the products that they say are formed, by means of micro-XRD, SEM-EDX, maybe TG-DSC could be very useful [2].

1.- doi:10.1016 / S0008-6223(03)00021-6

2.- doi.org/10.3390/ma13010098

Reviewer 3 Report

The overall performance of this article is not bad, but many issues must be revised or improved.

The grammar and content must be revised more. Your article is not a paper in the recent stage. You just label “1. Introduction”. The others are waived. The readers difficultly read it.

In Fig. 2(a) and (d) and Fig. 3(b), the words or labels are too blurry to read.

In the technical part, I have some concerns and you must step by step illustrate them.

1.      For the new coating material, SiO2+SiC+Al (H2PO4)3, the article topic talks about the mechanical and dielectric properties. However, these results are to insufficient to recognize the great performance in this article. You must aim on these points and concisely illustrate them.

2.      The dielectric characteristics of this coated film related to the electromagnetic (EM) wave absorption must be definitely demonstrated more. In abstract, the authors mentioned “Anti-oxidation mechanism investigations revealed that the coating effectiveness could be attributed to three substances, i.e. SiO2, SiP2O7 and SiO2 +AlPO4 solid solution from the reactions of SiC+O2SiO2+CO, SiO2+P2O5SiP2O7 and SiO2 +AlPO4 solid solution, respectively.” This good contribution is not distinct in the content.

3.      The electrical dielectric vs. EM wave frequency response is shown in Fig. 2(d). Because this factor is related to the transmission, absorption, and reflection of EM waves, I hope you should propose the measurement results and discuss their benefits on the three substances.

4.      Tables and theorems must be added or derived to verify your coating results.

5.      Most of the reference data are old. You must quote the newest development in this area.

Round 2

Reviewer 1 Report

Authors have incorporated all the queries in the manuscript.

Reviewer 2 Report

Dear authors,

Right now the main problem with the manuscript is that there is only one experiment. And this can never be a paper.

If the Materials journal has a section, like other journals, called letters, it could be accepted.

Reviewer 3 Report

Good revision.